# The Influence of Body Fat and Lean Mass on HbA1c and Lipid Profile in Children and Adolescents with Type 1 Diabetes Mellitus

**DOI:** 10.3390/diseases11040125

**Published:** 2023-09-23

**Authors:** Thais Menegucci, Eduardo Federighi Baisi Chagas, Barbara de Oliveira Zanuso, Karina Quesada, Jesselina Francisco dos Santos Haber, Tereza Laís Menegucci Zutin, Luis Felipe Pimenta, Adriano Cressoni Araújo, Elen Landgraf Guiguer, Claudia Rucco P. Detregiachi, Marcia Gabaldi Rocha, Patrícia Cincotto dos Santos Bueno, Lucas Fornari Laurindo, Sandra M. Barbalho

**Affiliations:** 1Postgraduate Program in Structural and Functional Interactions in Rehabilitation, School of Medicine, Universidade de Marília (UNIMAR), Marília 17525-902, São Paulo, Brazil; thaismenegucci@hotmail.com (T.M.);; 2Postgraduate Program of Health and Aging, School of Medicine, Faculdade de Medicina de Marília (FAMEMA), Marília 17519-030, São Paulo, Brazil; 3Interdisciplinary Center on Diabetes (CENID), Universidade de Marília (UNIMAR), Marília 17525-902, São Paulo, Brazil; 4Department of Biochemistry and Pharmacology, School of Medicine, Universidade de Marília (UNIMAR), Marília 17525-902, São Paulo, Brazillucasffffor@gmail.com (L.F.L.); 5Department of Biochemistry and Nutrition, School of Food and Technology of Marília (FATEC), Marília 17500-000, São Paulo, Brazil; 6Department of Animal Sciences, School of Veterinary Medicine, Universidade de Marília (UNIMAR), Marília 17525-902, São Paulo, Brazil; 7Department of Biochemistry and Pharmacology, School of Medicine, Faculdade de Medicina de Marília (FAMEMA), Marília 17519-030, São Paulo, Brazil

**Keywords:** type 1 diabetes mellitus, body fat, lean mass, glycated hemoglobin, apolipoprotein

## Abstract

Glycated hemoglobin (HbA1c) is used to assess glycemic control in Type 1 diabetes (DM1) patients. Apolipoproteins play an essential role in DM1 pathophysiology and may be associated with complications and HbA1c. This cross-sectional observational study of 81 children and adolescents of both sexes diagnosed with DM1 investigated the relationship between body fat distribution and lean mass with HbA1C and apolipoprotein values, analyzing biochemical and body composition measurements. A Shapiro–Wilk test with Lilliefors correction, a non-parametric Mann–Whitney test, and others were used with a significance level of 5%. The sample had a diagnosis time of 4.32 years and high blood glucose levels (mean 178.19 mg/dL) and HbA1c (mean 8.57%). Subjects also had a moderate level of adiposity, as indicated by arm and thigh fat areas. The study also found significant differences in the distribution of patients concerning levels of apolipoproteins A and B, with a smaller proportion of patients having undesirable levels. Finally, the study found a significant difference in the distribution of patients with estimated cardiovascular risk based on the ApoB/ApoA-I ratio. Conclusively, visceral fat in children and adolescents with DM1 may increase the risk of DM1 long-term complications owing to its association with elevated HbA1C and apolipoprotein values.

## 1. Introduction

Type 1 diabetes mellitus (DM1) is a very common chronic condition observed in childhood and adolescence that is related to autoimmune wrecking of pancreatic islets, resulting in partial or complete loss of ability to produce insulin, which can lead to severe complications if not adequately treated [1,2,3]. This condition has a significant economic and social impact in Brazil and the world. The losses are immeasurable, and the onset of the disease leads to a loss of quality of life. The change in social commitments brings negative results to children, adolescents, and their families [4,5].

DM1 is linked to genetics, environmental factors, and adaptive immunity. Initially, the disease includes chronic inflammation and degeneration of the β-cells in the pancreatic islets, leading to decreased insulin production. The inflammatory scenario is also related to DM1 complications such as diabetic nephropathy and retinopathy [6,7,8,9].

Although DM1 patients traditionally tend to be underweight, the increase in sedentary lifestyles and exacerbated consumption of foods high in sugar and fat has contributed to weight gain in this population. It is known that an increase in the proportion of fat, especially visceral fat, and a reduction in lean mass worsen glycemic control and plasma lipid levels [10,11,12]. In addition, the increase in visceral adipose tissue to the detriment of the reduction in lean mass has direct implications for installing a pro-inflammatory scenario and oxidative stress that will strongly contribute to the micro and macro-vascular complications [13,14,15,16].

Although intensive insulin therapy is crucial for glycemic control, it can also contribute to weight gain, primarily if performed improperly and with excessive calorie intake [17,18].

Glycated hemoglobin (HbA1c) is one of the main indicators of glycemic control and is strongly correlated with risks of long-term complications [17,19,20,21].

In addition to HbA1c, apolipoproteins (apo) A-I, A-II, and the Apo A-II/Apo A-I ratio contribute to the pathological process of DM1, playing a fundamental role in glucose metabolism and cardiovascular complications. Thus, alterations in these molecules can interfere with the glycemic control in DM1 and the proportion of adipose tissue. Studies show that lower levels of apo A or higher levels of apo B are correlated with signs of microvascular dysfunction, which are related to DM1 complications, such as the pathogenesis of diabetic retinopathy [22,23,24,25].

Although increased body fat is associated with worse metabolic control, there is evidence that different regions of fat deposition are associated with other metabolic consequences in terms of insulin sensitivity, serum lipids, adipokines, and inflammatory factors [26].

Therefore, assessing the distribution of lean mass and adipose tissue is essential to help control blood glucose and prevent future complications in DM1 patients [27]. Given the above, this study aimed to investigate the relationship between body fat and lean mass distribution on HbA1c, lipid profile, and Apo A-I/B values in children with DM1.

## 2. Materials and Methods

### 2.1. General Data

The sample size was calculated using the G*Power software, version 3.1.9.2 (Franz Faul, Universität of Kiel, Kiel, Germany), to analyze the association between overweight/obesity and apolipoprotein values in children and adolescents with DM1. Considering an expected proportion of 10% (0.10), a degree of freedom, and a large effect size (0.50), a minimum sample of 72 sample elements was estimated for a type I margin of error (α) of 1% and a study power of 95% [28].

The sample consisted of 81 children and adolescents of both sexes (59.3% male/40.7% female) diagnosed with DM1 for at least one and a maximum of 14 years and aged between 4 and 19 years. We collected the data from medical records from the database of the Medical Specialty Ambulatory in Associação Beneficente Hospital Universitário and in the Centro Interdisciplinar em Diabetes (CENID) from 2019 to 2020. Patients were referred by the Regional Health Department of Marília-DRS IX via the Health Service Supply Regulation Center (CROSS). The general project was approved by the Ethics and Research Committee of UNIMAR (opinion: 3,606,397/2019).

All patients with at least 12 months of DM1 diagnosis were included. CENID clinic serves patients aged 4 to 19 years, so the age range was large, which also reflects on the duration of the disease. However, the minimum disease duration was 12 months to avoid patients at the beginning of the condition. Controlling the effect of age and time of illness was considered in the exploratory correlation analysis and the regression model.

Patients who did not authorize access to the medical records by signing the Terms of Assent and Terms of Free and Informed Consent, had a diagnosis of Autistic Spectrum Disorder, or presented physical disability with malfunction or paralysis of upper and/or lower limbs, were not included in the study.

Patient data in this cross-sectional observational study were obtained by accessing clinical data filed in the CENID database. Data were collected on the patient’s clinical history (age, sex, time since diagnosis, therapeutic strategy, history of diseases and complications), pattern of habitual physical activity, body composition, lipid profile, fasting glucose, glycated hemoglobin (HbA1c) and apoliproteins A-I and B. Body composition was analyzed using anthropometric measurements of body mass, height, circumferences and skinfolds, as well as bioimpedance tests to estimate lean mass, percentage of fat and muscle mass. The biochemical tests used in the clinical routine of patients consist of fasting blood glucose, casual blood glucose, glycated hemoglobin (HbA1c), total cholesterol, LDL-cholesterol (LDL-c), HDL-cholesterol (HDL-c), triglycerides, VLDL-cholesterol (VLDL-c) and apolipoproteins “A” (ApoA-I) and “B” (ApoB). The glycated hemoglobin (HbA1c) dosage was performed using the high-performance liquid chromatography (HPLC) method. Fasting Glycemia, total cholesterol, HDL-c, and triglycerides were analyzed by the enzymatic colorimetric method. The dosage of ApoB and ApoA-I was performed by the nephelometry method. LDL-c was calculated using the Friedewald equation, and non-HDL-c was calculated using the total cholesterol—HDL-c.

Anthropometric measurements of body mass, height, skinfolds (triceps and medial thigh), and circumferences (thigh, arm, and waist) were performed following standardized recommendations [29]. From the anthropometric measurements, the conicity index was calculated [30], body mass index z-score (BMI z-score) [31], arm muscle area (AMB), arm area, arm fat area, and arm fat percentage [32], as well as thigh muscle area, thigh area, thigh fat area, and thigh fat percentage [33] for body composition analysis. The analysis of body composition was complemented with the estimate of the percentage of body fat (%F), body fat (kg), and lean mass percentage using the bioimpedance test (BIODYNAMICS BIA 310e equipment) [34].

### 2.2. Study Variables

The prevalence of chronic diseases in the population was obtained through a questionnaire of reported morbidities and confirmed by clinical diagnosis present in the medical referral and complemented with information about the time of diagnosis of the disease and information about the use of medications.

The analysis methods for the biochemical measurements were Glycemia, total cholesterol, HDL-c, triglycerides, and LDL-c by the enzymatic colorimetric method; HbA1c by high-performance liquid chromatography (HPLC); and ApoA-I and ApoB were performed by nephelometry or immunoturbidmetry. ApoA-I values below 120 (mg/dL) and ApoB above 90 (mg/dL) are not normal for children and adolescents. The following cutoff points were considered for stratifying the risk of acute myocardial infarction based on the values of the apo B/ApoA-I index for men and women, respectively: low risk 0.40–0.69/0.30–0.59; moderate risk 0.70–0.79/0.60–0.79; high risk 0.90–1.10/0.80–1.00 [35,36].

To analyze body composition, anthropometric measures of body mass, height, skinfolds, and circumferences were used, as well as body fat and lean mass data referring to the bioimpedance test. From the anthropometric measurements, body composition indicators were calculated.

Circumference, body weight, and height measurements were used to calculate the conicity index to analyze visceral fat [37]. For the estimation of lean mass and body fat through bioimpedance, specific equations for age and gender were used [38,39]. Arm circumference and triceps skinfold measurements were used to calculate arm fat area, arm muscle circumference, AMB, and AMBc [40,41]. Thigh circumference and thigh skinfold measurements were used to estimate thigh muscle and fat areas [42].

### 2.3. Data Analysis Methodology

Quantitative variables are described by mean, standard deviation (SD), and range (minimum and maximum value). Qualitative variables are described by absolute and relative frequency distribution. Differences in the proportion distribution of qualitative variables were analyzed using Fisher’s exact test. The normal distribution was verified using the Shapiro–Wilk test with Lilliefors correction. Spearman’s non-parametric correlation test was performed to explore the relationship between the ordinal quantitative and qualitative variables.

The multiple linear regression model was used to analyze the effect of body composition variables on the values of HbA1c, ApoA-I, ApoB, and ApoB/ApoA-I coefficient controlling the effect of sex, time of diagnosis, and pubertal stage by the Enter method. The R^2^ was analyzed to verify the determination coefficient of the variation percentage explained by the model. For all analyses, the SPSS software version 19.0 for Windows was used, with a significance level of 5%.

## 3. Results

Table 1 presents the mean, standard deviation, minimum, and maximum values of several clinical and anthropometric variables in a sample of individuals with DM1. In general, the mean values of the sample indicate that individuals have a relatively long time since diagnosis (4.32 years), with high levels of blood glucose (mean of 178.19 mg/dL) and HbA1c (mean of 8.57%). In addition, the average Apolipoprotein B/Apolipoprotein A-I ratio is greater than the value considered ideal for individuals at increased risk of cardiovascular disease. Furthermore, arm and thigh fat areas also indicate a moderate level of adiposity in the sample. As pointed out before, controlling the effect of age and duration of illness was considered in the exploratory correlation analysis and the regression model, but as these variables (age and time of illness) did not have a significant effect on the variables of interest, they will not be shown in the results.

Table 2 presents an analysis of variables that include sex, time of diagnosis, insulin administration, presence of associated morbidities, level of physical activity, level of glycated hemoglobin (HbA1c), levels of Apo A and B, cardiovascular risk, and index of body mass.

Regarding insulin administration, there was a significant difference (*p* < 0.001) in the proportion of patients using an insulin pump compared to those using a pen.

A significant difference was also found in the distribution of patients in the different HbA1c classes, considering that the proportion of patients with HbA1c above 8% was significantly higher compared to the other classes (*p* < 0.001).

There was a significant difference in the distribution of patients into ApoA-I and ApoB classes, as the proportion of patients with undesirable levels of these proteins was significantly lower compared to desirable levels (*p* < 0.001).

The table also shows a significant difference in the distribution of patients concerning the cardiovascular risk estimated by the ApoB/ApoA-I coefficient. Most patients were at low risk (*p* < 0.001). Finally, there was a significant difference in the distribution of patients in the different classes of BMIz. The proportion of thin/thin patients was significantly lower than the other classes (*p* < 0.001).

The other variables did not show statistically significant differences.

Table 3 presents a correlation analysis between various anthropometric and biochemical parameters concerning levels of HbA1c, ApoA-I, ApoB, and ApoB/ApoA-I coefficient. Correlation analysis was performed using Pearson’s correlation coefficient (r) and *p*-value.

The conicity index showed a positive and significant correlation with ApoB values (r = 0.227, *p*-value = 0.042*), indicating that the increase in visceral obesity is related to the increase in serum ApoB values. However, no correlation was found between the conicity index and ApoA-I and ApoB/ApoA-I index values. The lack of correlation between the conicity index and the ApoB/ApoA-I index was due to the lack of correlation with ApoA-I and the low correlation with ApoB.

The percentage of body fat measured by bioimpedance (Bio) did not show a significant positive correlation with the level of ApoA-I (r = 0.141, *p*-value = 0.210). Body fat (kg) measured by bioimpedance showed a significant positive correlation with HbA1c (r = 0.272, *p*-value = 0.014*). Lean mass (%) measured by bioimpedance showed a significant negative correlation with HbA1c (r = −0.275, *p*-value = 0.013*) and with the ApoB/ApoA-I coefficient (r = −0.104, *p*-value = 0.357).

In addition, some parameters, such as the BMI z-score, thigh muscle area, thigh area, thigh fat area, % arm fat, and arm fat area, did not show a significant correlation with any of the four parameters analyzed (HbA1c, ApoA-I, ApoB, and ApoB/ApoA-I coefficient).

In Table 4, a multiple linear regression analysis was performed to verify the effect of the body composition variables on HbA1c, ApoA-I, ApoB, and ApoB/ApoA-I coefficient, controlling for the variables sex, time of diagnosis, and pubertal stage. There was a significant effect of the percentage of fat and lean mass by bioimpedance on HbA1c values. The increase in the percentage of fat and the reduction in lean mass are related to the increase in HbA1c. The model did not significantly affect the fat and lean mass percentage. When analyzing the R^2^ value, it was verified that the percentage of fat and lean mass explains 6.4% and 8.4% of the HbA1c variation.

It was observed that the increase in AMB (cm^2^), controlling for the effect of gender, time of diagnosis, and pubertal stage, is related to the reduction in ApoB. This model significantly explains 13.0% (R^2^) of the ApoB variation. The AMB (cm2) increase is also associated with a decrease in the ApoB/ApoA-I coefficient. Although the model did not show a significant effect, R^2^ points out that the variation in AMB (cm^2^), controlling for the effect of gender, time of diagnosis, and pubertal stage, explains 10% of the variation in the ApoB/ApoA-I coefficient.

## 4. Discussion

Our study showed that the diagnosis time of the disease in the included patients was 4.32 years. High blood glucose levels had a mean of 178.19 mg/dL and HbA1c of 8.57%. Subjects also had a moderate level of adiposity, as indicated by arm and thigh fat areas. The study also showed significant differences in the distribution of patients concerning levels of apolipoproteins A and B, with a smaller proportion of patients with non-normal levels. Moreover, there was a difference in the distribution of patients with estimated cardiovascular risk based on the ApoB/ApoA-I ratio.

Considering the BMI Z-score, it was observed that most of the sample had a eutrophic nutritional status, but 22.2% were overweight and 2.5% obese. The sample also showed an inadequate glycemic control profile, as 54.3% had HbA1c greater than 8%. As for the cardiovascular risk profile analyzed by apolipoproteins, it was found that the largest proportion of the sample was at low risk, although 19.8% had undesirable ApoB values.

Although a large part of the sample showed adequate nutritional status, it was verified that the increase in body fat and the reduction in lean mass have a negative impact on glycemic control, with an increase in HbA1c. The apolipoprotein B profile followed anthropometric changes, and the increase in apo B was significantly associated with the increase in visceral fat analyzed by the conicity index. The degree of relationship between the conicity index and ApoB was low, considering the value of the correlation coefficient. Chen et al. [43] also obtained similar results and pointed out that ApoB is negatively correlated with lipolysis and positively correlated with visceral accumulation. However, the reduction in the arm muscle area negatively impacted ApoB and the ApoB/ApoA-I index.

Like our results, a Brazilian study with 120 DM1 children with a mean age of 11.47 also showed high mean values for HbA1c (8.13%). In this study, over 30% of the population was overweight [44]. In a survey carried out in Iran, Mostofizadeh et al. [45] showed that most of the studied DM1 individuals (n = 274 individuals under 19 years of age) had dyslipidemia and HbA1c over 8.3%.

A case–control study in Baghdad that aimed to investigate the nutritional status of children and adolescents with DM1 (mean age 10.0 ± 3.73 years in DM1 and 8.68 ± 3.1 in controls), showed that anthropometric measurements in DM1 patients were significantly lower than those in controls (*p* < 0.001). BMI z-score showed a significant negative correlation with HbA1c (r = −0.295, *p* = 0.006, respectively) [46].

The increase in the proportion of body fat (especially visceral fat) and the reduction in lean mass is negatively related to glycemic and lipid metabolism, leading to worse glycemic control and the need for insulin dose increments in DM1 patients [10,11,12]. Changes in insulin sensitivity are due to the change in the secretory pattern of adipose tissue from lean to obese, where M2 macrophages can be replaced by M1, resulting in an increased gene expression and production of pro-inflammatory mediators that are released by adipose visceral tissue. Among these mediators, we can mention Interleukin (IL) 6, Tumor Necrosis Factor-α (TNF-α), resistin, free fatty acids (FFA), and reduction in adiponectin and IL-10 levels, which have an anti-inflammatory action [13,14,15,16].

Muscle tissue has essential functions in maintaining homeostasis and is also linked to secretory actions. The resulting products are named myokines, cytokines, or growth factors that can play autocrine, paracrine, or endocrine roles. Among the numerous functions of these substances, it is possible to mention the improvement in glycemic control by reducing insulin resistance and improving protein and lipid metabolism. Several myokines have positive effects on glucose uptake and improvement in blood glucose. On the other hand, the pro-inflammatory scenario promoted by the reduction in the release of these myokines that, associated with poor glycemic control, leads to an increased risk of developing metabolic syndrome and cardiovascular complications [23,47,48,49].

Poor glycemic control, characterized by high concentrations of HbA1c, is one of the main clinical factors related to lipid alterations and consequent increased risk of micro and macrovascular complications. A case–control study designed to investigate risk factors for young DM1 individuals (10–22 years old) showed a negative influence on the lipid profile due to poor glycemic control. In addition, this study also showed that DM1 patients had significantly increased Apo B levels compared to non-diabetics, regardless of glycemic control [50].

Maahs et al. [51] studied children with DM1 (age at onset 10.6 ± 4.1 years and duration of DM1 around ten months) to investigate changes in HbA1c levels over a 2-year follow-up interval. They observed that changes in HbA1c over time were significantly linked to total cholesterol, LDL-c, HDL-c, and triglyceride levels, showing that improvement in glucose control was associated with a better lipid profile. Still, it was not sufficient to normalize lipid levels in young dyslipidemic type 1 diabetes.

The combination of dyslipidemia and DM1 accelerates atherogenesis, and young people and adults with this condition form a high-risk group in terms of cardiovascular disease. In DM children with adequate glycemic control, there are often no marked lipid abnormalities, but poor control increasingly deteriorates plasma lipid values, and their association with additional risk factors such as obesity, sedentary lifestyle, hypertension, smoking, or family history of heart disease may lead to premature development of atherosclerosis during adolescence [52].

The evaluation of apolipoprotein complement is critical for lipoprotein actions and metabolism. These large molecules can modulate enzyme actions or cellular receptors. Apo A-I is the primary lipoprotein in HDL-c and crucial for its anti-atherosclerotic effects, including reverse cholesterol transport from tissues to the liver. As in other autoimmune diseases, patients with DM1 can present dyslipidemia. Literature shows conflicting results concerning apo AI levels in DM1 patients compared with healthy controls. However, many authors show increased insulin levels can up-regulate apoA1 gene expression. Due to this reason, a rigid glycemic control is of utmost importance to reduce atherosclerotic issues in DM1 patients. Refs. [50,53,54,55] demonstrated that Apo AI levels can be significantly augmented in newly diagnosed DM1 subjects during the first twelve months of diagnosis, and the authors suggested that apo levels can work as markers of micro and macrovascular imbalance that are associated with the late DM1 complications.

Basu et al. [56] evaluated the relationship between the apolipoprotein profile and the occurrence of any cardiovascular event or major atherosclerotic cardiovascular events such as fatal or non-fatal myocardial infarction or stroke. During 15 years of follow-up, 50 events (defined as any cardiovascular event) showed a significant positive correlation with ApoB and other Apo.

In a prospective study, cardiovascular risk factors were evaluated in 175 children with DM1 who were compared with 150 non-diabetic children as controls. The results showed increased levels of pro-inflammatory biomarkers such as TNF-α, IL-4, and high-sensitivity C-reactive protein in patients with DM1. In addition, the presence of greater intima-media thickness of the artery and other cardiovascular risk factors were correlated with the time of diagnosis, increased BMI, levels of Apo A, ApoB, total cholesterol, and triglycerides. The authors concluded that the parameters mentioned above may be related to the early impairment of the structure and function of the common carotid and aortic arteries in young patients with DM1 [57]. The increase in pro-inflammatory cytokines and Apo B observed in this study may be related to the secretory pattern of visceral adipose tissue and the reduction in lean mass that can generally be observed in T1DM patients.

The increased vascular risk may, at least in part, be associated with the characteristic of LDL-c particles that are shown to be more susceptible to oxidation and more atherogenic than their larger counterparts; in addition, an increase in the proportion of small particles, representing profile B of the LDL-c pattern, seems to confer atherogenicity. This can be attributed to elevated triglyceride concentration, which increases VLDL production and impaired ability to clear these particles [45].

Due to the above, we can highlight the importance of monitoring blood glucose, plasma lipids, apolipoprotein levels, and body composition in DM1 patients to minimize future common complications and evaluate the impact of therapeutic interventions. Although the presented results demonstrate a low to moderate degree of relationship for the correlation coefficient, they are relevant for clinical practice since they are related to cardiovascular conditions.

## 5. Conclusions

Our results indicate that increased body fat and reduced lean body mass are related to poor glycemic control represented by HBA1c levels. On the other hand, alterations in Apo B and the Apo B/Apo A index were associated with alterations in the muscle tissue of the upper limbs. Although the observed relationships are low to moderate, the results suggest that more detailed body composition monitoring is necessary to minimize future complications in DM1 patients.

## Figures and Tables

**Table 1 diseases-11-00125-t001:** Descriptive statistics of the sample’s quantitative variables.

	Average	SD	Minimum	Maximum
Age (years)	12.60	3.58	4.00	19.00
Diagnostic time (years)	4.32	2.99	1.00	14.00
Glycemia (mg/dL)	178.19	69.14	73.00	429.00
Total cholesterol (mg/dL)	165.27	33.80	87.00	246.00
Triglycerides (mg/dL)	82.55	52.52	21.72	343.00
LDL-c (mg/dL)	89.37	27.41	24.00	171.00
HDL-c (mg/dL)	55.07	10.66	23.00	76.00
Non-HDL-c	110.20	34.43	37.00	201.00
HbA1c (%)	8.57	2.27	4.91	15.30
ApoA-I	149.00	17.89	95.00	213.00
ApoB	77.20	18.61	31.00	126.00
Apolipoprotein B/Apolipoprotein A-I coefficient	0.28	0.85	0.52	0.12
Conicity index	1.14	0.08	0.79	1.36
BMI z-score	0.25	1.23	−3.01	3.30
Arm muscle area (cm^2^)	20.93	3.15	15.37	28.72
Arm area (cm^2^)	59.75	11.09	40.75	95.00
Arm fat area (cm^2^)	12.36	2.84	7.87	22.98
Arm fat (%)	20.55	1.12	19.16	24.44
Thigh muscle area (cm^2^)	106.21	38.58	43.53	208.58
Thigh area (cm^2^)	157.18	58.38	66.96	336.39
Thigh fat area (cm^2^)	50.97	27.79	7.60	132.58
Thigh fat area (cm^2^) (%)	31.65	9.82	4.14	55.37
Fat bioimpedance (%)	21.73	7.63	9.60	40.00
Fat (kg) bioimpedance	10.93	6.01	2.42	28.20

SD, standard deviation.

**Table 2 diseases-11-00125-t002:** Absolute (*f*) and relative (%) frequency distribution of the qualitative variables that characterize the sample.

		*f*	%	*p*-Value
Gender	Male	48	59.3	0.119
	Female	33	40.7	
Diagnostic time class	<5 years	46	56.8	0.226
	>5 years	35	43.2	
Insulin administration	Bomb	22	27.2	<0.001 *
	Pen	59	72.8	
Associated morbidities	Yes	5	6.2	<0.001 *
	No	76	93.8	
Level of physical activity	Sedentary	47	58.0	0.182
	Little active	34	42.0	
HbA1c	<7%	20	24.7	<0.001 *
	7–8%	17	21.0	
	>8%	44	54.3	
ApoA-I	Desirable	79	97.5	<0.001 *
	Undesirable	2	2.5	
ApoB	Desirable	65	80.2	<0.001 *
	Undesirable	16	19.8	
Cardiovascular risk ApoB/ApoA-I coefficient	Low	71	87.7	<0.001 *
	Moderate	9	11.1	
	High	1	1.2	
Body Mass Index	Thinness/skinny	9	11.1	<0.001 *
	Eutrophic	52	64.2	
	Overweight	18	22.2	
	Obese	2	2.5	

* indicates a significant difference in the proportion distribution of response categories by Fisher’s exact test for *p*-value ≤ 0.050.

**Table 3 diseases-11-00125-t003:** Correlation analysis of HbA1c, ApoA-I, ApoB, and ApoB/ApoA-I coefficient with body composition variables.

	HbA1c (%)		ApoA-I		ApoB		ApoB/ApoA-I Coefficient	
	r	*p*-Value	r	*p*-Value	r	*p*-Value	r	*p*-Value
Conicity index	−0.086	0.447	0.039	0.730	0.227	0.042 *	0.119	0.289
z-score BMI	0.041	0.717	0.083	0.461	0.018	0.874	−0.085	0.449
Arm muscle area (cm^2^)	0.034	0.765	−0.172	0.126	−0.298	0.007 *	−0.269	0.015 *
Arm area (cm^2^)	0.034	0.761	−0.087	0.440	−0.148	0.186	−0.177	0.113
Arm fat area (cm^2^)	0.059	0.603	−0.066	0.561	−0.113	0.316	−0.152	0.175
arm fat (%)	0.116	0.303	0.059	0.602	0.140	0.213	0.071	0.531
Thigh muscle area (cm^2^)	0.098	0.383	−0.069	0.540	−0.195	0.081	−0.210	0.060
Thigh area (cm^2^)	0.137	0.224	−0.032	0.774	−0.118	0.296	−0.155	0.167
Thigh fat area (cm^2^)	0.170	0.129	0.010	0.928	0.071	0.527	0.010	0.926
Thigh fat area (cm^2^)	0.111	0.325	0.033	0.770	0.199	0.075	0.182	0.104
Fat bioimpedance (%)	0.252	0.023 *	0.141	0.210	0.181	0.106	0.076	0.499
Fat bioimpedance (kg)	0.272	0.014 *	0.088	0.435	0.091	0.417	−0.003	0.979
Lean mass bioimpedance (kg)	0.134	0.235	−0.105	0.351	−0.201	0.073	−0.190	0.090
Lean mass bioimpedance (%)	−0.275	0.013 *	−0.167	0.135	−0.211	0.059	−0.104	0.357

R, regression coefficient; * indicates significant correlation by Spearman’s test for *p*-value ≤ 0.050.

**Table 4 diseases-11-00125-t004:** Multiple linear regression analysis for the effect of body composition variables on HbA1c, ApoA-I, ApoB, and ApoB/ApoA-I coefficient controlling for gender, time of diagnosis, and pubertal stage.

Variables		B	CI 95%		*p*-Value	Model	
Dependent	Independent		LL	UL		*p*-Value	R^2^
HbA1c (%)	(Constant)	7.636	5.687	9.586	<0.001 *	0.274	0.064
	Gender	−0.539	−1.718	0.639	0.365		
	Diagnostic time (years)	−0.099	−0.297	0.098	0.321		
	Pubertal stage	0.072	−0.647	0.791	0.842		
	fat bioimpedance (%)	0.090	0.008	0.172	0.031 *		
HbA1c (%)	(Constant)	18.297	10.234	26.360	0.000	0.150	0.084
	Gender	−0.690	−1.878	0.498	0.251		
	Diagnostic time (years)	−0.107	−0.303	0.088	0.278		
	Pubertal stage	0.068	−0.641	0.778	0.849		
	Lean mass bioimpedance (%)	−0.108	−0.192	−0.024	0.013 *		
ApoB	(Constant)	124.241	86.233	162.250	0.000	0.030 **	0.130
	Gender	−1.783	−11.201	7.636	0.707		
	Diagnostic time (years)	−0.394	−1.976	1.188	0.621		
	Pubertal stage	7.926	0.578	15.274	0.035		
	Arm muscle area (cm^2^)	−2.855	−4.796	−0.913	0.004 *		
ApoB/ApoA-I	(Constant)	0.839	0.583	1.095	0.000	0.088	0.100
	Gender	−0.047	−0.111	0.016	0.141		
	Diagnostic time (years)	−0.004	−0.014	0.007	0.495		
	Pubertal stage	0.047	−0.003	0.096	0.065		
	Arm muscle area (cm^2^)	−0.016	−0.029	−0.003	0.017 *		

B, regression coefficient; CI 95%, confidence interval of 95%; LL, lower limit; UP, upper limit; gender (1 = male; 2 = female); pubertal stage (1 = pre-pubertal; 2 = pubertal; 3 = post-pubertal); * indicates significant effect of the independent variable; ** indicates significant model effect; coefficient of determination of the percentage of variation explained by the model (R^2^).

## Data Availability

No applicable.

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
