# Peer review of "The Influence of Body Fat and Lean Mass on HbA1c and Lipid Profile in Children and Adolescents with Type 1 Diabetes Mellitus"

_diseases, 2023, doi:10.3390/diseases11040125_

Round 1
Reviewer 1 Report
I read your manuscript, I find it interesting, and it represents an advancement in the area. Although it could be published in its actual form, I suggest improving the description of your results it will make them more accessible to a broad audience. Also, try to focus the discussion on your results, and reinforce your discussion with the conclusions from similar published studies.
Author Response
I read your manuscript, I find it interesting, and it represents an advancement in the area. Although it could be published in its actual form, I suggest improving the description of your results it will make them more accessible to a broad audience. Also, try to focus the discussion on your results, and reinforce your discussion with the conclusions from similar published studies.
Response: Dear doctor. Thank you very much for your time reviewing this manuscript.
Response to the comment: We improved the description of the results. Please, see lines 173-225.
Reviewer 2 Report
Manuscript can be accepted with rechecking of grammer and language
Manuscript can be accepted with rechecking of grammer and language
Author Response
Manuscript can be accepted with rechecking of grammer and language.
Response: Dear doctor. Thank you very much for your time reviewing this manuscript.
The text was reviewed by a native speaker.
Reviewer 3 Report
The aim of this study was to investigate the relationship between body fat and lean mass distribution on HbA1C, lipid profile and Apo A/B values in children with DM1. Therefore the title should be changed in line with the aim pursued.
In the introduction, I do not find much justification for the need to conduct this study, because the described relationships are known. The authors themselves mention that: assessing the distribution of lean mass and adipose tissue is important to help control blood glucose, contributing to the prevention of future complications in DM1 patients (lines 78-80). The introduction requires improvement and showing the justification for the conducted research in a group of children and adolescents with DM1 and showing the logical whole, not individual citations.
Materials and Methods - what inclusion and exclusion criteria were used? why was there such a big discrepancy in age and disease duration? I have not found information whether this has been taken into account in further analyses. The description of the methodology is imprecise - lines 108-111: body size and body composition is not the same; which circumferences, skinfolds were measured? who carried out the measurements and according to what procedures? (give adeqate references); what equipment was used? were the measurements repeated? What conditions for BIA measurement have been maintained?
lines 111-114 vs. 120-123? - give one precise description for the parameters used.
"The analysis methods for the biochemical measurements were Glycemia.." - it is incomprehensible.
Why did you use AMI and adult cutoffs? On the other hand, I have not found results for this.
lines 131-139 - which indicators, equations, formulas were used? what reference values/interpretation criteria were used for them.
Results
Table 1 - Taper index, BMI z-score - were not included in the methods; lean mass - were included in the methods , no in table
Apolipoprotein B / Apolipoprotein A-I coefficient vs. Cardiovascular risk ApoB/ApoA-1 coefficient
No explanations for the abbreviations used and no consistency in their use.
BMI class - please add adequate references which were used.
It may be worth showing the results broken down into children with and without controlled glycemia.
Discussion - in the first paragraph, show the main results related to the title and the aim. The way of writing the discussion is substantively incorrect, the obtained results have not been discussed, only the individual works of other authors have been cited. This requires thorough improvement so as to highlight particular problems in the light of existing knowledge. I am missing a word in the discussion regarding the impact of age, developmental stage, sex, duration of the disease on changes/differences in body composition in the study group.
Author Response
- The aim of this study was to investigate the relationship between body fat and lean mass distribution on HbA1C, lipid profile and Apo A/B values in children with DM1. Therefore the title should be changed in line with the aim pursued.
Response: Dear doctor. Thank you very much for your time reviewing this manuscript. We modified the title for Influence of Body Fat and Lean Mass on HbA1c and Lipid Profile in Children and Adolescents With Type 1 Diabetes Mellitus.
- In the introduction, I do not find much justification for the need to conduct this study, because the described relationships are known. The authors themselves mention that: assessing the distribution of lean mass and adipose tissue is important to help control blood glucose, contributing to the prevention of future complications in DM1 patients (lines 78-80). The introduction requires improvement and showing the justification for the conducted research in a group of children and adolescents with DM1 and showing the logical whole, not individual citations.
Response: Dear doctor, we believe that the sentence in lines 78-82 can justify your concern. Moreover, we included a reference that show the importance of the lipid profile in the glycemic control in T1DM patients.
- Materials and Methods - what inclusion and exclusion criteria were used? why was there such a big discrepancy in age and disease duration? I have not found information whether this has been taken into account in further analyses. The description of the methodology is imprecise - lines 108-111: body size and body composition is not the same; which circumferences, skinfolds were measured? who carried out the measurements and according to what procedures? (give adeqate references); what equipment was used? were the measurements repeated? What conditions for BIA measurement have been maintained?
Response: Thank you for this suggestion. We improved the description of the Methods section. Please see in lines 120-136. Moreover, we improved the results section. Please see lines 182-187, and 221-226 (and tables). Moreover, Anthropometric measurements of body mass, height, skinfolds (triceps and medial thigh) and circumferences (thigh, arm and waist) were performed following standardized recommendations (ACSM, 2011). From the anthropometric measurements, the conicity index was calculated (Calcaterra et al., 2022), body mass index z-score (BMI z-score) (Onis et al., 2007), arm muscle area ( AMB), arm area (AB), arm fat area (AGB) and arm fat % (%GB) (Frisancho & Tracer, 1987), as well as thigh muscle area (AMC), thigh area (AC), thigh fat area (AGC) and % thigh fat (%BF) (Housh et al., 1995) for body composition analysis. The analysis of body composition was complemented with the estimate of the percentage of body fat (%F), body fat (kg), lean mass (%) and percentage of lean mass using the bioimpedance test (BIODYNAMICS BIA 310e equipment) (Mundstock et al., 2021). Please, see the new references that were included to respond to your suggestion:
ACSM. (2011). ACSM’s manual for evaluation of physical fitness and health (Editora Guanabara Koogan (ed.)).
Calcaterra, V., Biganzoli, G., Ferraro, S., Verduci, E., Rossi, V., Vizzuso, S., Bosetti, A., Borsani, B., Biganzoli, E., & Zuccotti, G. (2022). A Multivariate Analysis of “Metabolic Phenotype” Patterns in Children and Adolescents with Obesity for the Early Stratification of Patients at Risk of Metabolic Syndrome. Journal of Clinical Medicine, 11(7), 1856. https://doi.org/10.3390/jcm11071856
Frisancho, A. R., & Tracer, D. P. (1987). Standards of arm muscle by stature for the assessment of nutritional status of children. American Journal of Physical Anthropology, 73(4), 459–465. https://doi.org/10.1002/ajpa.1330730408
Housh, D. J., Housh, T. j, Weir, J. p, Wier, L. L., Johnson, G. o, & Stout, J. R. (1995). Amthropometric estimation of thigh muscle cross-setional area. Medicine and Science in Sports and Exercise, 784–791.
Mundstock, E., Vendrusculo, F. M., Filho, A. D., & Mattiello, R. (2021). Consuming a low-calorie amount of routine food and drink does not affect bioimpedance body fat percentage in healthy individuals. Nutrition, 91–92, 111426. https://doi.org/10.1016/j.nut.2021.111426
Onis, M. De, Onyango, A. W., Borghi, E., Siyam, A., Nishida, C., & Siekmann, J. (2007). Development of a WHO growth reference for school-aged children and adolescents. Bulletin of the World Health Organization, 85(09), 660–667. https://doi.org/10.2471/BLT.07.043497
- Lines 111-114 vs. 120-123? - give one precise description for the parameters used.
Response: Dear doctor, thank you for this suggestion. We included the precise description. Please, see lines 120-134.
- "The analysis methods for the biochemical measurements were Glycemia.." - it is incomprehensible.
Response: Dear doctor, the description was included in lines 120-134.
- Why did you use AMI and adult cutoffs? On the other hand, I have not found results for this.
Response: The results are shown in Table 2. The cut-off points used can also be used in the study population as there is no standardization of cut-off points for the age group of the study.
- lines 131-139 - which indicators, equations, formulas were used? what reference values/interpretation criteria were used for them.
Response: Dear doctor. We understand your concern. However, the equations or formulas can be accessed by consulting the cited references, therefore, we understand that it is not necessary to insert the formulas in the text of the article. The only parameter that was categorized was the BMI z-score, but this was not described because its cutoff points were already established in the literature and contained in the aforementioned reference.
- Results
Table 1 - Taper index, BMI z-score - were not included in the methods; lean mass - were included in the methods , no in table.
Response: Dear reviewer, both the conicity (not taper) index and the BMI z-score are described in the method. Lean mass was only presented in Table 3.
Apolipoprotein B / Apolipoprotein A-I coefficient vs. Cardiovascular risk ApoB/ApoA-1 coefficient.
Response: Dear reviewer, they are the same.
No explanations for the abbreviations used and no consistency in their use.
Response: Dear doctor, all the abbreviations were revised.
BMI class - please add adequate references which were used.
Response: Dear reviewer, this explanation was included in the text according to your comments above.
It may be worth showing the results broken down into children with and without controlled glycemia.
Response: Dear reviewer, this relationship has already been explored within the regression.
Discussion - in the first paragraph, show the main results related to the title and the aim. The way of writing the discussion is substantively incorrect, the obtained results have not been discussed, only the individual works of other authors have been cited. This requires thorough improvement so as to highlight particular problems in the light of existing knowledge. I am missing a word in the discussion regarding the impact of age, developmental stage, sex, duration of the disease on changes/differences in body composition in the study group.
Response: Dear reviewer, thank you for this suggestion. We proceeded as you suggested.
Dear doctor, we know your time is precious. Thank you again for reviewing this manuscript. With regards.
Reviewer 4 Report
This paper reveals significant differences in the distribution of patients with type I diabetes mellitus with estimated cardiovascular risk based on the ApoB/ApoA-1 ratio.
These findings in patients with type I diabetes are few and rare and of great interest.
However, several issues can be pointed out.
The authors need to analyze and discuss apolipoproteins in more detail in order to describe their association with apolipoproteins. For example, it may be necessary to examine ApoA-I and ApoB by dividing them into quartiles, or to analyze the ApoB/LDL-C ratio, the ApoA-I/HDL-C ratio, or the relationship with treatment duration and patient background.
In this paper, the descriptions of ApoA and ApoA-I are mixed; ApoA has both ApoA-I and ApoA-II, which should be clarified. If ApoA is ApoA-I+ApoA-II, that should be carefully stated.
Does the description "undesireble" for ApoA-I and ApoB mean outside the normal range?
page 6; line 194, "The results show that the conicity index had a significant correlation with the ApoB level (r = 0.227, p-value = 0.042*) and the ApoB/ApoA -I coefficient (r = 0.119, p-value = 0.289)." This result describes a mistake.
Page 6; line 194, "The percentage of body fat measured by bioimpedance (Bio) showed a significant 198 positive correlation with the level of ApoA (r = 0.141, p-value = 0.210)." This result likewise describes a mistake.
Page 6; line 201," Lean mass (%) measured by bioimpedance showed a significant negative correlation with HbA1c (r = -0.275, p-value = 0.013 *) and with the ApoB/ApoA-I coefficient (r = -0.104, p-value = 0.357). " In this last sentence, the sentence, and the content, should be revised because the association with the ApoB/ApoA-I coefficient significantly indicates a poor correlation.
The authors should clearly describe how all blood parameters such as apolipoproteins and LDL cholesterol are measured; they should also clarify whether LDL-C is measured by a direct method.
Author Response
This paper reveals significant differences in the distribution of patients with type I diabetes mellitus with estimated cardiovascular risk based on the ApoB/ApoA-1 ratio.
These findings in patients with type I diabetes are few and rare and of great interest.
Response: Dear doctor. Thank you very much for your time reviewing this manuscript.
However, several issues can be pointed out.
The authors need to analyze and discuss apolipoproteins in more detail in order to describe their association with apolipoproteins. For example, it may be necessary to examine ApoA-I and ApoB by dividing them into quartiles, or to analyze the ApoB/LDL-C ratio, the ApoA-I/HDL-C ratio, or the relationship with treatment duration and patient background.
Response: Dear doctor, thank you for this comment. The sample size is not appropriate for exploring quartile subdivisions. The disease duration and the characteristics of the patients were already considered in the regression model, however the variables without significant effect were not presented.
In this paper, the descriptions of ApoA and ApoA-I are mixed; ApoA has both ApoA-I and ApoA-II, which should be clarified. If ApoA is ApoA-I+ApoA-II, that should be carefully stated.
Does the description "undesireble" for ApoA-I and ApoB mean outside the normal range?
Response: Dear reviewer, we did the corrections. Thank you for this comment. And yes, undesirable means not normal.
page 6; line 194, "The results show that the conicity index had a significant correlation with the ApoB level (r = 0.227, p-value = 0.042*) and the ApoB/ApoA -I coefficient (r = 0.119, p-value = 0.289)." This result describes a mistake.
Response: Dear doctor, we understand your concern, however, we do not consider it a mistake. First, because ApoB and ApoB/Apoa-I can correlate with each other, which is expected, it does not mean that they are the same variable and that they will present the same degree of relationship with the conicity index. Furthermore, the fact that the ApoB/ApoA-I index does not correlate with the conicity index can be explained by the low correlation of ApoB with the conicity index and the lack of correlation between ApoA-I and the conicity index.
Page 6; line 194, "The percentage of body fat measured by bioimpedance (Bio) showed a significant 198 positive correlation with the level of ApoA (r = 0.141, p-value = 0.210)." This result likewise describes a mistake.
Response: Aqui o revisor está correto. Temos que corrigir o texto, pois, não existe correlação significtiva
Page 6; line 201," Lean mass (%) measured by bioimpedance showed a significant negative correlation with HbA1c (r = -0.275, p-value = 0.013 *) and with the ApoB/ApoA-I coefficient (r = -0.104, p-value = 0.357). " In this last sentence, the sentence, and the content, should be revised because the association with the ApoB/ApoA-I coefficient significantly indicates a poor correlation.
Response: Dear reviewer, you are completely right. We correct the text. Please see lines 230-231.
The authors should clearly describe how all blood parameters such as apolipoproteins and LDL cholesterol are measured; they should also clarify whether LDL-C is measured by a direct method.
Response: You are right again. We included these descriptions in the methods section. Please, see lines 121-143.
Dear doctor, we know your time is precious. Thank you again for reviewing this manuscript. With regards.
Reviewer 5 Report
Menegucci et al., have studied the association of body composition with HbA1C/Apolipoprotein in children and adolescents with type 1 diabetes. Although it’s relevant to study the influence of fat mass and lean mass on the metabolic risk of developing diabetic complications and cardiovascular diseases, there are some major concerns.
1. In this study, body composition (body fat and lean body mass) was assessed by bioimpedance, specific equations for age and gender were used. Nevertheless, it is the 4-compartment model (4C) (DXA) which is regarded as the reference method in body composition. Therefore, authors need to discuss the accuracy and reliability of this method.
2. Since body composition and AMB measurements in this study may have high variabilities, given a small sample size of 88 (although power was calculated), a replication study is warranted.
3. Is the increased body fat/reduced lean body mass the cause or effect of poor glycemic control represented by HBA1c levels?
Minor edits in English.
Author Response
Menegucci et al., have studied the association of body composition with HbA1C/Apolipoprotein in children and adolescents with type 1 diabetes. Although it’s relevant to study the influence of fat mass and lean mass on the metabolic risk of developing diabetic complications and cardiovascular diseases, there are some major concerns.
Response: Dear doctor. Thank you very much for your time reviewing this manuscript.
- In this study, body composition (body fat and lean body mass) was assessed by bioimpedance, specific equations for age and gender were used. Nevertheless, it is the 4-compartment model (4C) (DXA) which is regarded as the reference method in body composition. Therefore, authors need to discuss the accuracy and reliability of this method.
Response: Dear reviewer. We understand your concern. However, the method available during the research period was bioimpedance and a reference was cited for the equation used. Although DXA can provide more accurate body composition data, the bioimpedance method is accepted by the literature, as well as the equations used.
- Since body composition and AMB measurements in this study may have high variabilities, given a small sample size of 88 (although power was calculated), a replication study is warranted.
Response: Dear doctor. We can understand your concern. In fact, a larger sample can have a significant impact on the results, but at the moment we cannot change this. However, new patients have been included in the outpatient care routine and in a future study we will be able to bring new results or those that confirm the findings in this study.
- Is the increased body fat/reduced lean body mass the cause or effect of poor glycemic control represented by HBA1c levels?
Response: Increased body fat and reduced lean body mass may contribute to poorer glycemic control. However, in the same way, high levels of glycemia can contribute to loss of lean mass and, consequently, an increase in the percentage of fat.
Dear doctor, we know your time is precious. We appreciate your comments and suggestions to improve this manuscript.
Round 2
Reviewer 3 Report
Thank you for taking into account most of the comments, but in my opinion the discussion still needs improvement. The main results obtained should be subjected to a logical discussion with the observance of thematic paragraphs
Author Response
Thank you for taking into account most of the comments, but in my opinion the discussion still needs improvement. The main results obtained should be subjected to a logical discussion with the observance of thematic paragraphs. Why don't the authors try to analyze and discuss apolipoproteins in more detail, as the reviewers pointed out? It would be advisable to explore this point further.
Dear Doctor, thank you very much for this suggestion. We improved the Discussion section. Please see lines 268-276; 327-332, and 333-343.
Reviewer 4 Report
Why don't the authors try to analyze and discuss apolipoproteins in more detail, as the reviewers pointed out? It would be advisable to explore this point further.
Page 6; line 194, "The results show that the conicity index was significantly correlated with ApoB levels (r = 0.227, p-value = 0.042*) and ApoB/ApoA-I coefficient (r = 0.119, p-value = 0.289)". The authors reply that they do not consider this to be an error, but the conicity index clearly does not correlate significantly with ApoB/ApoA-I.
It should be added that the correlation coefficient between the conicity index and ApoB levels, r = 0.227, is also a very low correlation and should be discussed in this light.
Author Response
Comments:
Page 6; line 194, "The results show that the conicity index was significantly correlated with ApoB levels (r = 0.227, p-value = 0.042*) and ApoB/ApoA-I coefficient (r = 0.119, p-value = 0.289)". The authors reply that they do not consider this to be an error, but the conicity index clearly does not correlate significantly with ApoB/ApoA-I.
It should be added that the correlation coefficient between the conicity index and ApoB levels, r = 0.227, is also a very low correlation and should be discussed in this light.
Dear Doctor, thank you very much for this suggestion. We improved the manuscript according to your suggestions. Please see lines 206-222:
Table 3 presents a correlation analysis between various anthropometric and biochemical parameters concerning levels of HbA1c, ApoA-I, ApoB, and ApoB/ApoA-I coefficient. Correlation analysis was performed using Pearson's correlation coefficient (r) and p-value.
The conicity index showed a positive and significant correlation with ApoB values (r = 0.227, p-value = 0.042*), indicating that the increase in visceral obesity is related to the increase in serum ApoB values. However, no correlation was found between the conicity index and ApoA-I and ApoB/ApoA-I index values. The lack of correlation between the conicity index and the ApoB/ApoA-I index was due to the lack of correlation with ApoA-I and the low correlation with ApoB.
The percentage of body fat measured by bioimpedance (Bio) did not show a significant positive correlation with the level of ApoA-I (r = 0.141, p-value = 0.210). Body fat (kg) measured by bioimpedance showed a significant positive correlation with HbA1c (r = 0.272, p-value = 0.014*). Lean mass (%) measured by bioimpedance showed a significant negative correlation with HbA1c (r = -0.275, p-value = 0.013*) and with the ApoB/ApoA-I coefficient (r = -0.104, p-value = 0.357).
Dear reviewer, thank you again for your time reviewing our manuscript. With best regards.
Reviewer 5 Report
- Authors have responded to the comments.
Minor editing of English Language.
Author Response
Minor editing of English Language.
Dear Doctor, thank you very much for this comment. We re-checked the manuscript for the language.